# The Antimicrobial Activity of Cannabinoids

**DOI:** 10.3390/antibiotics9070406

**Published:** 2020-07-13

**Authors:** John A. Karas, Labell J. M. Wong, Olivia K. A. Paulin, Amna C. Mazeh, Maytham H. Hussein, Jian Li, Tony Velkov

**Affiliations:** 1Department of Pharmacology and Therapeutics, The University of Melbourne, Melbourne 3010, VIC, Australia; jkaras@unimelb.edu.au (J.A.K.); labellw@student.unimelb.edu.au (L.J.M.W.); oliviakapaulin@gmail.com (O.K.A.P.); amna.mazeh@unimelb.edu.au (A.C.M.); maytham.hussein@unimelb.edu.au (M.H.H.); 2Biomedicine Discovery Institute and Department of Microbiology, Monash University, Melbourne 3800, VIC, Australia; jian.li@monash.edu

**Keywords:** antibiotics, antimicrobial resistance, MRSA, *Cannabis sativa*, cannabinoids, structure–activity relationships

## Abstract

A post-antibiotic world is fast becoming a reality, given the rapid emergence of pathogens that are resistant to current drugs. Therefore, there is an urgent need to discover new classes of potent antimicrobial agents with novel modes of action. *Cannabis sativa* is an herbaceous plant that has been used for millennia for medicinal and recreational purposes. Its bioactivity is largely due to a class of compounds known as cannabinoids. Recently, these natural products and their analogs have been screened for their antimicrobial properties, in the quest to discover new anti-infective agents. This paper seeks to review the research to date on cannabinoids in this context, including an analysis of structure–activity relationships. It is hoped that it will stimulate further interest in this important issue.

## 1. Introduction

The discovery of antibiotics is undoubtedly one of the most important scientific achievements in modern medicine and has saved millions of lives since their discovery. However, their misuse has resulted in the emergence of antibacterial resistance (AMR), which is a significant threat to global human health. It is estimated that global deaths could reach 10 million per year by 2050 with a cumulative cost to global economic output of US$100 trillion [1]. The World Health Organization has highlighted that AMR is one of the three greatest threats to healthcare systems globally [2]. For example, immuno-compromised patients such as organ transplant recipients and those who are undergoing chemotherapy treatment are particularly vulnerable and depend on the efficacy of antibiotics to fight off infection [3,4]. The spread of multidrug-resistant (MDR) bacteria therefore threaten these significant advancements in medical care. Despite this, only four new classes of antibiotics have reached the clinic in the past four decades [5]. The development of new classes of anti-infective agents with novel modes of action is therefore essential to address the growing threat of AMR.

*Cannabis sativa* is an herbaceous plant that belongs to the family Cannabinaceae [6]. It has been used by humans for over 5,000 years for medicinal and recreational uses, firstly in Central and Northeast Asia and subsequently spreading worldwide [7]. Cannabis, or marijuana, is undoubtedly one of the most widely used illicit drugs [8]. It has a complex chemical composition that includes cannabinoids [9], which are a group of secondary metabolites, several of which are responsible for the psychotropic effects [10,11]. Metabolic fingerprinting of *C. sativa* unveiled more than 480 compounds, including many terpenes and approximately 180 cannabinoids [12]. Their biosynthesis involves the alkylation of olivetolic acid with geranyl diphosphate, which is catalyzed via geranyl transferase [13]. This reaction leads to the formation of cannabigerolic acid (CBGA, Figure 1A), which is a precursor molecule for numerous other cannabinoids, such as Δ^9^-tetrahydrocannbinol (Δ^9^-THC), cannabidiol (CBD), cannabigerol (CBG), cannabinol (CBN) and cannabichromene (CBC) (see Figure 1B–F) [14,15].

Cannabinoids bind to cannabinoid receptors CB_1_ and CB_2_ in humans [11], and are distributed throughout the body, including in immune cells. It has been suggested that activation of these receptors could modulate the function of the immune system during infection [16]. At least 113 different cannabinoids have been isolated from *C. sativa*, many of which exhibit various pharmacological effects [17]. One of the most studied and main intoxicating cannabinoids is Δ^9^-THC, which exerts its effects through partial agonism at the CB_1_ receptor [18,19]. Another major constituent is CBD, but this compound is not intoxicating [18]. Most cannabinoids are metabolized in the liver by cytochrome P450 [20].

Although the medicinal use of cannabis in Europe dates back to the 13th century, it was only popularized in the 19th century when it was found to have anticonvulsive, analgesic, antianxiety and antiemetic effects [21]. Over the past decade, advocates of medicinal cannabis have highlighted its potential for treating a variety of conditions, including cancer [22]. This has sparked a range of reactions, from reluctance and caution [23] to excitement [24]. The reluctance stems from the adverse effects of cannabis, which limit its widespread therapeutic use. Cannabis use has been known to cause psychological effects for at least 4,000 years [25], including symptoms such as euphoria, intensification of sensory perceptions and impaired motor skills in healthy individuals [26]. Most commonly, it has well-documented deleterious effects on neurocognitive functioning in humans [27]. It has also been linked to deficits in working and episodic memory [28], executive functions [29], anhedonia and anxiety [30]. A recent report by the WHO also suggested an association between myocardial infarction or stroke with heavy cannabis use [31]. The adverse effects on the cardiovascular system is most likely caused by CB_1_ receptor activation by Δ^9^-THC, while CBD seems to have beneficial effects [32]. Moreover, some medical professionals have concerns about prescribing medicinal cannabis (where it is legalized) due to its potential addictive properties. However, it should be acknowledged that there are disparities and contradictions in findings between studies. Regardless, healthcare practitioners should be aware of these adverse effects and inform their patients before prescription. Recently, evidence surrounding the use of cannabinoids on treatment-resistant refractory epilepsy has increased [33,34,35]. There are also strong indications that cannabinoids are effective agents for the treatment of chronic pain, nausea post-chemotherapy and appetite stimulation [36,37,38,39,40].

## 2. Antibacterial Effects of *C. sativa* Extracts

The first reports detailing the antibacterial activity of cannabinoids date back to the 1950s [41,42]. These experiments were conducted before the phytochemistry of cannabis was well characterized, which means that the bactericidal effect of *C. sativa* could not be directly attributed to a specific constituent. This was achieved in 1976, where it was found that ∆^9^-THC and CBD are bacteriostatic as well as bactericidal against a panel of Gram-positive pathogens (see below) [43]. There has also been great interest in the antibacterial properties of the essential oils and various extracts from *C. sativa*, such as those derived from petroleum ether, methanol and hot water. Various methods for isolating *C. sativa* extracts have also been applied. Traditional techniques include cold-pressing and solvent extraction, however higher yielding technologies which generate superior products are now emerging [44]. Pressurized liquid extraction circumvents the need for filtration, whereas ultrasonic-assisted extraction (UAE) methods use less solvent and have short processing times, with higher yields and equivalent quality. “Green” methods include microwave-assisted extraction and supercritical fluid extraction, however up-scaling is challenging for the latter process [44].

Novak and co-workers evaluated the essential oils of five different cultivars of *C. sativa* against a large panel of Gram-positive and Gram-negative pathogens. The most abundant compounds in each oil sample were α-pinene, myrcene, *trans*-β-ocimene, α-terpinolene, *trans*-caryophyllene and α-humulene. The antimicrobial activity was in general poor, with only modest activity against *Acinetobacter calcoaceticus* and *Brevibacterium linens*. No ∆^9^-THC and very low levels of CBD and CBN were detected in all the essential oils, which suggests a role of these compounds in the antimicrobial activity of *C. sativa* [45]. A later study evaluated the oil of the seeds from the whole plant, extracted by petroleum ether and methanol. It was found (via the cup–plate agar diffusion method) that each extract exhibited an antimicrobial effect against Gram-positive pathogens. This is consistent with the study by Van Klingeren et al. [43], although the cannabinoid content of the plant specimen was not characterized. No significant antifungal activity was observed. Intriguingly, minor activity against Gram-negative bacteria was also reported, albeit a modest effect (note that the petroleum ether extract was inactive against *Pseudomonas aeruginosa*) [46]. Hot water and ethanol leaf extracts have also been shown to possess an inhibitory effect against Gram-negative pathogens [6].

Lone and Lone investigated and compared the protein yield and antimicrobial and antioxidant effects of *C. sativa* following both aqueous and acetone extraction [47]. From 500 g samples, protein extraction from aqueous methodologies yielded 4.8 g of crude extract and 3.2 g from acetone. When compared with the aqueous extract, the acetone extract displayed fractionally superior bactericidal properties. A concentration-dependent response was observed against all strains, with *V. cholera* being marginally the most responsive bacteria, closely followed by *P. aeruginosa*. The fungal response was slightly more pronounced in *C. albicans* compared to *C. neoforms*. Additionally, this study found that *C. sativa* has antioxidant properties, thus widening its potential for clinical use.

Sarmadyan et al. investigated the antimicrobial properties of “Hashish” against common hospital-associated bacterial strains [48]. Disk diffusion experiments found that cannabis extract exerted the greatest antimicrobial effects on *S. aureus* 25923, with an inhibition zone of 14 mm, followed by MRSA, *E. coli* and *K. pneumoniae* with values of 12, 10 and 7 mm, respectively. No activity was observed in *P. aeruginosa* or *A. baumannii*. This trend regarding spectrum of activity is consistent with the findings of Vu et al., whereby extracts from Vietnamese derived *C. sativa* plants were found to possess modest antimicrobial activity against Gram-positive bacteria, with Gram-negative bacteria being less susceptible [49]. Similarly, Lelario and co-workers also observed that the major components of *C. sativa* extracts displayed moderate activity only against Gram-positive pathogens [50].

Hemp seed oil–water emulsions have also been under investigation for pharmaceutical and cosmetic applications, due to their low toxicity and biodegradability. The antimicrobial activity of two emulsions, consisting of both unrefined and refined oil, was determined. Overall, the bactericidal effect against Gram-positive pathogens was modest and less so for Gram-negative bacteria (virtually zero activity against *Escherichia coli* CCM 3954 was observed). The refined oil was also less potent than the unrefined oil. This could be due to the higher content of α-linolenic acid in the latter, or more likely the fact that ∆^9^-THC is removed during the refinement process [51]. Ethanol extracts of *C. sativa* leaves have also been tested against both clinical samples and nonclinical MRSA isolates, via the disc diffusion method. For the clinical isolates, the inhibition zone diameter ranged 9–15 mm, which was somewhat less than vancomycin (13–24 mm). However, 1:1 combination with other medicinal plant extracts such as *Psidium guajava* and *Thuja orientalis* demonstrated a synergistic effect, whereby inhibition zone diameters ranged 25–30 mm in most cases. This synergy may be due to the presence of quercetin, gallic acid and flavonoids such as catechin, which were detected in the leaf extracts; however, no cannabinoids were detected [52]. Interestingly, Frassinetti and co-workers observed inhibitory activity of *C. sativa* seed extracts on *S. aureus* biofilm formation, which indicates that these extracts could have enormous potential as preservatives in both the food and cosmetics industries [53]. Similarly, Stahl et al. found that cannabinoids are more effective in reducing the bacterial colony count in dental plaque when compared with commercial toothpastes such as Oral B and Colgate, which suggests that *C. sativa*-derived compounds could be used for oral care applications [54].

Investigation is also underway into the biological activity of commercially viable ∆^9^-THC-free essential oil of *C. sativa*, which could have medicinal, cosmetic, veterinary, agronomic or food applications. The oil was evaluated against several *S. aureus* strains and exhibited both a moderate antibacterial effect and antibiofilm activity, which was partly attributable to the flavanone naringenin. Antimicrobial activity was also observed against *Helicobacter pylori* (a Gram-negative organism) but no anti-fungal activity was observed. This study demonstrates that *C. sativa* is a rich source of biologically active compounds and that its antimicrobial properties are not solely attributable to cannabinoids [55]. Hemp seed hexane extracts have also been evaluated. The resultant oil was found to inactivate the growth of acne-causing *Propionibacterium acnes* as well as possess an anti-inflammatory effect [56].

Recently, Iseppi and co-workers conducted an extensive phytochemical characterization of 17 hemp essential oils from different varieties. Seventy-one compounds were identified, with the terpenes β-myrcene, α-pinene, α-terpinolene, β-pinene, *trans*-ocimene and limonene being the most abundant. CBDV, CBC and CBD were also detected, but below 0.05% of the percentage area in each chromatogram. The minimum inhibitory concentration (MIC) values of each oil were determined against a panel of Gram-positive pathogens and were found to have moderate to good antimicrobial activity. To determine the source of the observed antibacterial effect, the afore-mentioned terpenes and CBD were also evaluated. In general, good to moderate activity was observed against *Listeria monocytogenes* and *Enterococcus* isolates, but slightly less so for *Staphylococcus* and *Bacillus* isolates. Therefore, the antimicrobial effect is likely to be due to synergism between several compounds that are present in the essential oils [57].

An interesting application for *C. sativa*-derived compounds is to use them for water filtration and purification purposes. Nadir et al. immobilized a mixture of cannabinoids and terpenes onto a polyethersulfone hybrid membrane. Bacterial decline was observed for both Gram-positive and Gram-negative pathogens. For example, an 88% decrease bacterial load for *S. aureus* was measured for the hybrid membrane compared to 30% for the standard membrane. Similar results were reported for several other species, including for *Pseudomonas aeruginosa* and *Escherichia coli*. This study represents a cost-effective solution for water filtration and antibacterial purification [58]. All these examples highlight the great potential of *C. sativa* and its antimicrobial properties for applications in food, agriculture and drug discovery.

## 3. Structure–Activity Relationships of Cannabinoids

Although it has long been known that marijuana has antibacterial properties, its potential to address the major global crisis of antibiotic resistance is largely untapped. There have been a few studies, however, which are outlined below. When reading this section, please refer to Figure 2 and the Appendix A, which contain each numbered structure that is indicated in bold text.

A study by van Klingeren et al. found that both ∆^9^-THC and CBD were active against several *S. aureus* and *Streptococcus* isolates, with MICs in the order of 1–5 μg/mL (**1**, **2**) [43]. However, this bactericidal effect was attenuated in the presence of horse serum, possibly due to the cannabinoids binding to plasma proteins. Only poor antimicrobial activity against Gram-negative pathogens was observed. Turner and co-workers evaluated the antibacterial and antifungal properties of a panel of cannabichromene analogs [59]. The *n*-pentyl chain meta to the alcohol group appears to be important for activity against *B. subtilis* and *S. aureus* (**3**–**5**). Interestingly, truncation to a methyl group appeared to enhance antifungal activity. Isocannabichromenes were also investigated (**6**, **7**) but were slightly less active than their analogous cannabichromene. This class of cannabinoid does not induce intoxicating effects, which improves their therapeutic potential.

Appendino et al. investigated the antibacterial activity of five major cannabinoids and their analogs against a range of MDR *S. aureus* isolates [60]. It was found that cannabidiolic acid (CBDA) possesses good antimicrobial activity (MIC = 2 μg/mL, **8**), which was further improved with omission of the carboxylate moiety (**9**). This has recently been verified by Martinenghi and co-workers, who observed that against several Gram-positive pathogens, CBD had MIC values in range of 1–2 μg/mL, compared with CBDA (2–4 μg/mL) [61]. The methyl and phenethyl esters of CBDA were inactive (**13**, **14**), which could be caused by the added hydrophobicity and/or steric bulk. Acetylation or methylation of the hydroxyl groups (**10**–**12**) was also detrimental for activity and indicates the importance of the resorcinol moiety. The MIC for cannabichromene (**15**) was 1–2 μg/mL, which is consistent with the findings of Turner et al. [59]. CBGA (**16**) possessed moderate activity, which was again improved with removal of the carboxylate (CBG, **17**). Acetylation (**18**) and methylation (**19**, **20**) of the hydroxyls was again detrimental for activity, as was esterification of CBGA (**21**, **22**). ∆^9^-THC acid (**23**) possessed only a moderate bactericidal effect compared to ∆^9^-THC and CBN (**24, 25**). Interestingly, swapping the hydroxyl at position 1 and the *n*-pentyl group at position 3 in both CBD and CBG (**26**, **27**) did not significantly affect activity. Di-hydroxylation of the prenyl tail in CBG (**28**) attenuated potency; prenylation at position 2 abolished activity. Resorcinol was also evaluated with and without an *n*-pentyl group (**30**, **31**); although the overall antimicrobial effect was poor, it still demonstrated the importance of the hydrocarbon chain.

Recently, Feldman et al. evaluated the endocannabinoids anandamide and arachidonyl serine (**32**, **33**) [62]. Although they exhibited poor bactericidal activity against planktonic MRSA isolates, they strongly inhibited biofilm formation, with a reduction of metabolic activity of pre-formed biofilms. Furthermore, cell aggregation, hydrophobicity, membrane potential and spreading ability, which are biofilm-associated virulence determinants, were altered. Therefore, these agents have potential as a treatment for recalcitrant MRSA biofilm infections, potentially in combination with existing antibiotics such as ampicillin and gentamicin [63]. This approach is effective, as it has been demonstrated that CBD (**34**) potentiates the antimicrobial effect of the peptide drug bacitracin against *Staphylococcus* species, *L. monocytogenes* and *E. faecalis* [64].

Farha and co-workers evaluated several cannabinoid analogs against MRSA USA300 and *E. coli* [65]. Several common cannabinoids (**35**–**43**, **45**, **47**) demonstrated moderate to good activity; these results mostly align with previous studies (vide supra). Δ^9^-Tetrahydrocannabivarin (**44**) and related analogs (**46**, **48**, **49**), which bear a common *n*-propyl chain at position 3, experienced up to a four-fold increase in MIC values, thus further highlighting the importance of the *n*-pentyl chain, which could play a role in membrane insertion. Carboxylation (**50**) and hydroxylation (**51**) at position 11 of Δ^9^-THC resulted in a loss of activity which indicates that lipophilicity may be important in the prenyl tail. Polycyclic cannabicyclol had low activity (**52**). CBG showed promising levels of efficacy in a murine systemic infection model of MRSA whereby the bacterial burden in the spleen was reduced by a factor of 2.8 log_10_ in CFU. None of these analogs displayed a bactericidal effect against *E. coli* however, with consistent MIC values of >128 μg/mL recorded. Interestingly, CBG was found to be effective against Gram-negative organisms in combination with polymyxin B or the less nephrotoxic polymyxin B nonapeptide. It is proposed that polymyxins permeabilize the outer membrane of Gram-negative pathogens to enable CBG to act on the inner membrane. Similarly, Kosgodage et al. found that CBD can act as a sensitizing agent in combination with various antibiotics [66]. For Gram-negative pathogens, CBD strongly inhibited the release of membrane vesicles, which play a role in inter-bacterial communication and the transfer of cargo molecules. When *E. coli* VCS257 was treated with CBD in combination with erythromycin, vancomycin, rifampicin, kanamycin or colistin, an enhanced antimicrobial effect was observed. These results are significant as they highlight the potential of cannabinoids as potentiators to improve the efficacy and broad-spectrum activity of existing antibiotic drugs.

## 4. Conclusions

*C. sativa* is a plant with untapped potential. It has an extensive metabolic profile and its medicinal properties should not be overlooked or overshadowed by its overuse as a recreational drug. Current antibiotic drugs have limited efficacy against MDR bacteria and their usage can be limited due to their toxicity for prolonged treatments; therefore, the discovery of an antimicrobial therapy of plant origin will no doubt be a great advancement in the field of anti-infectives [6]. Several cannabinoids have been found to have potent antimicrobial activity against Gram-positive pathogens such as MRSA isolates. Endocannabinoids have also been shown to be effective in eradicating biofilms. Combination therapy with bactericidal agents that possess different modes of action such as polymyxin B have shown synergism and broad-spectrum activity. There is also evidence that other compounds found in *C. sativa* such as terpenes have promising antimicrobial activity, which warrants further investigation. As bacteria are rapidly developing resistance against existing drugs, cannabinoids present a novel and exciting opportunity as a potential new source of antibiotics.

## Figures and Tables

**Figure 1 antibiotics-09-00406-f001:**
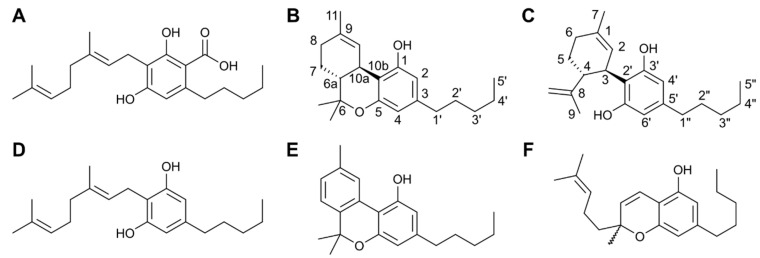
Chemical structures of the most common cannabinoids: (**A**) CBGA; (**B**) Δ9-THC, with the numbering convention included; (**C**) CBD, also numbered; (**D**) CBG; (**E**) CBN; and (**F**) CBC.

**Figure 2 antibiotics-09-00406-f002:**
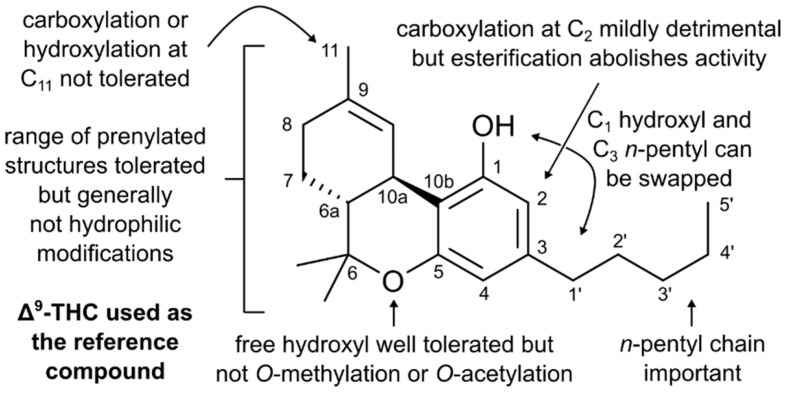
Summary of cannabinoid structure–activity relationships. Please note that this is a rough guide and the conclusions drawn therein are only relevant to antimicrobial activity against Gram-positive pathogens such as *S. aureus.*

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
