# Peer review of "The Antimicrobial Activity of Cannabinoids"

_antibiotics, 2020, doi:10.3390/antibiotics9070406_

Round 1

Reviewer 1 Report

This review article (antibiotics-842826) describes the antimicrobial activity of Cannabinoids. The article is very superficially written and does not go into the details of each study. Authors need to expand on their discussion and incorporate more figures and tables to make it a comprehensive reading. Additionally, a very similar and more comprehensive article (Antibiotics 20209(6), 297; https://doi.org/10.3390/antibiotics9060297) was recently (2 June 2020) published in the same journal. Hence, this review article will not be of any additional value for the readers of ANTIBIOTICS. Hence, I would reject and encourage authors to revise and submit into another journal for consideration. 

Author Response

Reviewer Comments

Reviewer 1

This review article (antibiotics-842826) describes the antimicrobial activity of Cannabinoids. The article is very superficially written and does not go into the details of each study. Authors need to expand on their discussion and incorporate more figures and tables to make it a comprehensive reading. Additionally, a very similar and more comprehensive article (Antibiotics 2020, 9(6), 297; https://doi.org/10.3390/antibiotics9060297) was recently (2 June 2020) published in the same journal. Hence, this review article will not be of any additional value for the readers of ANTIBIOTICS. Hence, I would reject and encourage authors to revise and submit into another journal for consideration.

We feel that our manuscript provides a detailed and unique perspective on the structure-activity relationships of cannabinoids.

We have also included a commentary on several other studies and have improved readability.

Finally, we have cited Klahn's excellent review.

Reviewer 2 Report

I placed my opinion in a pdf file.

Author Response

The manuscript presented for revision is very interesting. This work concerns a very important area of science, especially now. Analysis of the antimicrobial activity properties of various plants, including Cannabis sativa L., is very important today. The manuscript discusses many papers published in recent years. This work is well organized and scientifically sound.

I suggest including in this manuscript the research presented in the publications:

  1. Hernández-Cervantes R. et al., Immunoregulatory Role of Cannabinoids during Infectious Disease. Neuroimmunomodulation 2017, 24, 183–199. https://www.karger.com/Article/FullText/481824
  2. ZhengHai, Z., Yan, D., YanRu, J., QingLi, Y., & ZhenWei, L. (2019). Antibacterial activity and stability of extract from hemp (Cannabis sativa L.) leaves. Journal of Food Safety and Quality, 10(4), 927-933.
  3. Frassinetti, S., Gabriele, M., Moccia, E., Longo, V., & Di Gioia, D. (2020). Antimicrobial and antibiofilm activity of Cannabis sativa L. seeds extract against Staphylococcus aureus and growth effects on probiotic Lactobacillus spp. LWT, 124, 109149.

We have included a discussion on each of the suggested manuscripts (except for ZhengHai et al., which is written in mandarin) and cited them accordingly.

Reviewer 3 Report

The Authors provide a comprehensive review based on scientific reports and new findings regarding the antimicrobial activity of cannabinoids. The Authors are encouraged to have a native English speaker go through this manuscript. The manuscript requires extensive English editing. Some examples that need correction are:

Line 18: analysis “of” structure-activity

Line 19: interest “in” this important issue

Line 81: inform their patients “before” prescription

Line 127: “in” most cases

Line 146: against “a” panel of

Line 192: with “the” removal

Line 237: the discovery of antimicrobial

In the text:

  • Cannabis sativa (and C. sativa): italic
  • bacterial species: italic

Furthermore, the Authors did not report several interesting works on the same topic:

  • Stahl, V., & Vasudevan, K. (2020). Comparison of Efficacy of Cannabinoids versus Commercial Oral Care Products in Reducing Bacterial Content from Dental Plaque: A Preliminary Observation. Cureus12(1).
  • Kosgodage, U. S., Matewele, P., Awamaria, B., Kraev, I., Warde, P., Mastroianni, G., ... & Lange, S. (2019). Cannabidiol is a novel modulator of bacterial membrane vesicles. Frontiers in cellular and infection microbiology9, 324.
  • Fathordoobady, F., Singh, A., Kitts, D. D., & Pratap Singh, A. (2019). Hemp (Cannabis sativa L.) extract: Anti-microbial properties, methods of extraction, and potential oral delivery. Food Reviews International35(7), 664-684.
  • Lelario, F., Scrano, L., De Franchi, S., Bonomo, M. G., Salzano, G., Milan, S., ... & Bufo, S. A. (2018). Identification and antimicrobial activity of most representative secondary metabolites from different plant species. Chemical and Biological Technologies in Agriculture5(1), 13.
  • Hernández-Cervantes, R., Méndez-Díaz, M., Prospéro-García, Ó., & Morales-Montor, J. (2017). Immunoregulatory role of cannabinoids during infectious disease. Neuroimmunomodulation24(4-5), 183-199.
  • Vu, T. T., Kim, H., Tran, V. K., Le Dang, Q., Nguyen, H. T., Kim, H., ... & Kim, J. C. (2015). In vitro antibacterial activity of selected medicinal plants traditionally used in Vietnam against human pathogenic bacteria. BMC complementary and alternative medicine16(1), 32.
  • Sarmadyan, H., Solhi, H., Najarian-Araghi, N., & Ghaznavi-Rad, E. (2014). Determination of the Antimicrobial Effects of Hydro-Alcoholic Extract‎ of Cannabis Sativa on Multiple Drug Resistant Bacteria Isolated from‎ Nosocomial Infections. Iranian Journal of Toxicology7(23), 967-972.
  • Lone, T. A., & Lone, R. A. (2012). Extraction of cannabinoids from Cannabis sativa L. plant and its potential antimicrobial activity. Universal Journal of Medicine and Dentistry1(4), 51-55.

Author Response

The Authors provide a comprehensive review based on scientific reports and new findings regarding the antimicrobial activity of cannabinoids. The Authors are encouraged to have a native English speaker go through this manuscript. The manuscript requires extensive English editing. Some examples that need correction are:

Line 18: analysis “of” structure-activity. Done.

Line 19: interest “in” this important issue. Done.

Line 81: inform their patients “before” prescription. Done.

Line 127: “in” most cases. Done.

Line 146: against “a” panel of. Done.

Line 192: with “the” removal. We prefer the existing wording.

Line 237: the discovery of antimicrobial. We prefer the existing wording.

In the text:

Cannabis sativa (and C. sativa): italic. All instances are already italicized.

bacterial species: italic. All instances are already italicized.

Furthermore, the Authors did not report several interesting works on the same topic:

Stahl, V., & Vasudevan, K. (2020). Comparison of Efficacy of Cannabinoids versus Commercial Oral Care Products in Reducing Bacterial Content from Dental Plaque: A Preliminary Observation. Cureus, 12(1).

Kosgodage, U. S., Matewele, P., Awamaria, B., Kraev, I., Warde, P., Mastroianni, G., ... & Lange, S. (2019). Cannabidiol is a novel modulator of bacterial membrane vesicles. Frontiers in cellular and infection microbiology, 9, 324.

Fathordoobady, F., Singh, A., Kitts, D. D., & Pratap Singh, A. (2019). Hemp (Cannabis sativa L.) extract: Anti-microbial properties, methods of extraction, and potential oral delivery. Food Reviews International, 35(7), 664-684.

Lelario, F., Scrano, L., De Franchi, S., Bonomo, M. G., Salzano, G., Milan, S., ... & Bufo, S. A. (2018). Identification and antimicrobial activity of most representative secondary metabolites from different plant species. Chemical and Biological Technologies in Agriculture, 5(1), 13.

Hernández-Cervantes, R., Méndez-Díaz, M., Prospéro-García, Ó., & Morales-Montor, J. (2017). Immunoregulatory role of cannabinoids during infectious disease. Neuroimmunomodulation, 24(4-5), 183-199.

Vu, T. T., Kim, H., Tran, V. K., Le Dang, Q., Nguyen, H. T., Kim, H., ... & Kim, J. C. (2015). In vitro antibacterial activity of selected medicinal plants traditionally used in Vietnam against human pathogenic bacteria. BMC complementary and alternative medicine, 16(1), 32.

Sarmadyan, H., Solhi, H., Najarian-Araghi, N., & Ghaznavi-Rad, E. (2014). Determination of the Antimicrobial Effects of Hydro-Alcoholic Extract‎ of Cannabis Sativa on Multiple Drug Resistant Bacteria Isolated from‎ Nosocomial Infections. Iranian Journal of Toxicology, 7(23), 967-972.

Lone, T. A., & Lone, R. A. (2012). Extraction of cannabinoids from Cannabis sativa L. plant and its potential antimicrobial activity. Universal Journal of Medicine and Dentistry, 1(4), 51-55.

We have included a discussion on each of the suggested manuscripts and cited them accordingly.

Round 2

Reviewer 1 Report

Following the incorporation of revisions requested by all three reviewers, it can be accepted for publication. 

Reviewer 3 Report

The Authors fully implemented the manuscript. I have no further comments.